# Relative Dose Intensity of Induction-Phase Pazopanib Treatment of Soft Tissue Sarcoma: Its Relationship with Prognoses of Pazopanib Responders

**DOI:** 10.3390/jcm8010060

**Published:** 2019-01-08

**Authors:** Kenji Nakano, Yuki Funauchi, Keiko Hayakawa, Taisuke Tanizawa, Keisuke Ae, Seiichi Matsumoto, Shunji Takahashi

**Affiliations:** 1Department of Medical Oncology, The Cancer Institute Hospital, Japanese Foundation for Cancer Research, Tokyo 135-8550, Japan; s.takahashi-cheomotherapy@jfcr.or.jp; 2Department of Orthopedic Surgical Oncology, The Cancer Institute Hospital, Japanese Foundation for Cancer Research, Tokyo 135-8550, Japan; yuki.funauchi@jfcr.or.jp (Y.F.); keiko.hayakawa@jfcr.or.jp (K.H.); taisuke.tanizawa@jfcr.or.jp (T.T.); keisuke.ae@jfcr.or.jp (K.A.); smatsumoto@jfcr.or.jp (S.M.)

**Keywords:** soft tissue sarcoma, tyrosine kinase inhibitor, pazopanib, relative dose intensity

## Abstract

The approved standard dose of pazopanib is 800 mg per day, but the appropriate dose of pazopanib to treat soft tissue sarcoma (STS) patients in real-world practice is controversial. Of 124 STS patients treated with pazopanib, we retrospectively analyzed the cases of STS patients who achieved progression-free survival at 12 weeks by pazopanib treatment as pazopanib responders, and we evaluated their relative dose intensity (RDI) in the initial 12 weeks (12W-RDI). We enrolled 78 STS patients in the analyses as pazopanib responders, and 54 patients of the 78 pazopanib responders (69%) were able to maintain 12W-RDI ≥80%. In landmark analyses, patients with 12W-RDI of 80% ≥80% had significantly longer progression-free survival compared to those with 12W-RDI <80% (30.7 weeks vs. 22.0 weeks, hazard ratio [HR]: 0.56 [95%CI: 0.33–0.94], *p* = 0.026). The most frequently observed reasons of treatment interruption and/or dose reduction of pazopanib during the initial 12 weeks were anorexia and liver function disorders. Liver toxicity was the adverse event most frequently observed in the 12W-RDI <80% patients throughout the treatment periods. Based on our results, it appears that maintaining as high a dose intensity as possible that is tolerable—at least during the initial 12 weeks—is likely to be the better option in pazopanib treatment for STS patients.

## 1. Introduction

Pazopanib is an oral multi-target tyrosine kinase inhibitor that is known to inhibit vascular endothelial growth factor receptor (VEGFR), platelet-derived growth factor receptor (PDGFR), fibroblast growth factor receptor (FGFR), c-kit, and other growth factors [1]. Based on the randomized clinical trial EORTC 62072 and PALETTE study [2], pazopanib was approved for the treatment of soft tissue sarcoma (STS); it is the first molecular-targeted drug to be used for the treatment of STS other than gastrointestinal stromal tumors (GISTs).

Based on the evaluation of feasibility and tolerability in a phase 1 clinical trial [3], the recommended phase 2 dose of pazopanib was set as 800 mg per day, and the dose became the approved standard dose; however, not all STS patients tolerate this standard dose. By an integral analysis of prospective clinical trials, the median dose intensity of pazopanib was estimated as 591 mg/day [4], or approx. 75% of the standard dose. There is also a clinical report proposing that a reduced dose of pazopanib could be more beneficial for STS patients [5].

Pazopanib is also approved for the treatment of renal cell carcinoma (RCC) [6,7]. The standard dose of pazopanib for RCC is the same as that for STS, and analyses of clinical responses and plasma pazopanib concentrations suggested that the pazopanib exposure was related to the RCC patients’ responses [8]. However, in an exposure-survival analysis of pazopanib, the relationship between the serum concentration of pazopanib and patients’ responses was not as strong in STS as it was RCC [9].

There is quite a difference between STS and RCC in the heterogeneity of pathological diagnoses. More than 50 histological subtypes of STS are covered in the World Health Organization (WHO) classification [10]. In a clinical trial of STS, responses to pazopanib differed when histological subtypes were different [11], which affected the approval of pazopanib to some STS histology, such as liposarcoma. This heterogeneity makes it difficult to evaluate the relationship between the dose intensity or exposure of pazopanib and pazopanib’s efficacy in STS patients, but planning studies based on each histology would be more difficult; STSs account for only 5% of all malignant diseases, and they include many rare histological types (detected in <100 cases worldwide per year) [10].

For evaluations of the relationship between pazopanib exposure and efficacy in STS in a heterogeneous patient population, it is necessary to extract populations in which the differences of treatment exposure are more strongly related to efficacy than the differences of histology. In general, the longer the total treatment period becomes, the greater the influence of treatment content efficacy (e.g., dose intensity) would be expected. We thus conducted the present study to evaluate the relationship between pazopanib dose density and its efficacy in STS patients by enrolling STS patients who showed clinical responses to pazopanib.

## 2. Material and Methods

### 2.1. Patients and Treatments

We retrospectively reviewed the cases of 124 STS patients treated with pazopanib at the Cancer Institute Hospital of the Japanese Foundation for Cancer Research during the period from December 2012 to March 2018. Pazopanib was initially prescribed as 800 mg per day, except for patients with some baseline organ disorders, such as liver dysfunction; 95 of them (77%) began on pazopanib as a form of outpatient treatment. Patients were booked to visit the hospital and receive physical examinations as well as a lab data check every 1–4 weeks, and evaluation of responses by CT scan and/or MRI were performed every 6–8 weeks. Dose reduction or treatment interruption in the case of an adverse event was allowed based on the treating physician’s decision. Among the 124 STS patients, those who achieved 12 weeks of progression-free survival (12W-PFS), which we defined as “pazopanib responders”, were enrolled in the analyses. We defined the threshold of pazopanib response as 12W-PFS based on the traditional outcome in a phase 2 trial of STS, including pazopanib [11,12].

### 2.2. Analyses

For the pazopanib responders, we evaluated the relative dose intensity (RDI) of pazopanib during the initial 12 weeks of pazopanib treatment (12W-RDI), and we evaluated the patients’ long-term prognoses. The RDI was calculated as the actual daily dose of pazopanib received by the patients divided by the standard daily dose (800 mg). We used 80% as the threshold of 12W-RDI; based on the results of landmark analyses, we compared the PFS and overall survival (OS) from the landmark period that was defined as 12 weeks after the pazopanib induction between pazopanib responders with 12W-RDI ≥80% and those with 12W-RDI <80%.

The PFS and OS were evaluated by the Kaplan-Meier method, and the differences in prognoses by 12W-RDI were compared by a log-rank test. The hazard ratios [HRs] were evaluated by a Cox regression analysis. In all analyses, *p*-values <0.05 were considered significant. SPSS ver. 25.0 software was used for the statistical analyses. The disease progression was evaluated by CT scan and/or MRI based on RECIST ver. 1.1. Adverse events during pazopanib treatment were evaluated based on the CTCAE ver. 4.01. This retrospective study was approved by our hospital’s institutional review board (IRB) (No. 2018-1020).

## 3. Results

### 3.1. Enrolled Patients in the Analyses

With the median follow-up duration of 38.3 weeks (range: 3.4–234.9 weeks), the median PFS of the 124 pazopanib-treated STS patients was 17.4 weeks (95%CI 14.4–20.5) (Figure 1); of these, the 78 patients (63%) who achieved 12W-PFS were included in the analyses as pazopanib responders.

Objective responses were observed in nine of 124 patients (nine partial response, objective response rate was 7.2%). All patients who showed objective responses were included as pazopanib responders. All of their best responses were shown during the initial 12 weeks of pazopanib treatment.

The patient characteristics of all of the STS patients and the pazopanib responders are summarized in Table 1. There were no significant differences in the patients’ age, gender, performance status (PS), primary lesions (extremities or not), histology (leiomyosarcoma, synovial sarcoma, liposarcoma, and other histology), or the number of previous systemic therapies (≤1 regimen or ≥2 regimens) between these two groups.

### 3.2. Relative Dose Intensity (RDI) of Pazopanib

RDI of the 124 pazopanib-treated STS patients throughout pazopanib treatment periods are shown in Figure 2; 67 of 124 patients (54%) achieved 100% of RDI, and 91 patients (73%) maintained a RDI of ≥80% throughout their treatment periods.

The pazopanib treatment doses of 78 pazopanib responders at the initial 12 weeks are shown in Figure 3. The median 12-RDI of pazopanib was 100% (range 13–100%); 42 of the 78 pazopanib responders (54%) maintained a 12W-RDI at 100%, and 54 of the 78 (69%) maintained a 12W-RDI of ≥80%.

The most frequently observed reasons for treatment interruption and/or dose reduction during the initial 12 weeks in the group of pazopanib responders were anorexia and liver function disorders; both were observed in 11 patients. Fatigue (eight patients), thrombocytopenia (four), neutropenia (three), hand-foot syndrome (three), diarrhea (two), and pneumothorax (two) were also observed as reasons for treatment interruption and/or dose reduction.

The median RDI of pazopanib responders through all the treatment periods was 88% (range 12–100); the median RDI of patients with 12W-RDI ≥80% was 98% (range 56–100), and those of patients with 12W-RDI <80% was 66% (range 12–81).

### 3.3. The Prognoses of the Pazopanib Responders

The PFS and OS of all 124 STS patients compared by RDI throughout the pazopanib treatment are shown in Figure 4. From a comparison of RDI throughout the treatment periods, patients with RDI ≥80% had significantly shorter PFS and OS compared to those with RDI <80% (median PFS was 13.6 weeks vs. 22.4 months, *p* = 0.027; median OS was 40.4 weeks vs. 77.0 weeks, *p* = 0.019).

On the other hand, in the landmark analyses, the PFS and OS of the pazopanib responders were 26.7 weeks (95%CI 19.9–33.5) and 71.0 weeks (95%CI 54.6–87.4), respectively. The comparison of the 12W-RDI of pazopanib revealed that the patients with 12W-RDI ≥80% had significantly longer PFS compared to those with 12W-RDI <80% (median 30.7 weeks vs. 22.0 weeks, HR 0.56, 95%CI 0.33–0.94, *p* = 0.026); however, there was no significant difference in OS between these two groups (median 71.0 weeks vs. 70.9 weeks, HR 0.88, 95%CI 0.44–1.76, *p* = 0.71) (Figure 5).

### 3.4. Adverse Events Observed in Pazopanib Responders

The adverse events observed during all treatment periods in the pazopanib responders are listed in Table 2.

Adverse events observed in >10% of the 12W-RDI <80% patients were lymphocytopenia, anemia, thrombocytopenia, fatigue, increased total bilirubin, increased AST (aspartate transaminase)/ALT (alanine transaminase), increased serum creatinine, and electrolytes disorders. Anorexia, diarrhea, pneumothorax, and hypothyroidism were observed in >10% of the 12W-RDI ≥80% patients. Regarding adverse events of Grade 3 or more, increased AST/ALT was highly observed in the 12W-RDI <80% group (33%).

## 4. Discussion

Most oral tyrosine kinase inhibitors approved for malignant diseases are prescribed with a fixed dose, regardless of the patient’s body surface area or body weight; however, there are exceptions to this, such as lenvatinib treatment for hepatocellular carcinoma (HCC) [13]. The interruption or dose reduction of a drug will reduce the dose intensity, and thus result in a reduction of the drug’s efficacy. In chronic myeloid leukemia patients, the dose intensity of tyrosine kinase inhibitors was reported to be associated with treatment responses, although there were differences in the clinical impact of the drugs [14,15]. In cases of solid tumors, a relationship between the dose intensity and prognoses was shown in RCC patients treated with sunitinib [16,17,18] and sorafenib [19].

For pazopanib, 800 mg/day is the approved standard dose for both STS and RCC. However, due to adverse events, not all patients can tolerate the standard dose, and it has been suggested that a reduced dose of pazopanib could be more beneficial to patients [5]. Multicenter retrospective data in clinical practice showed that the median dose intensity of pazopanib administered to STS patients was 609–712 mg/day [20,21], which was the same as or higher than the median dose intensity observed in clinical trials (591 mg/day) [4]. In RCC, there was a clinical trial that evaluated a reduced dose of pazopanib in an adjuvant setting, and the trial did not show clinical efficacy [22]. In metastatic RCC, a reduced dose of pazopanib was reported to be related to relatively poor outcomes [23]. Whether the approved pazopanib dose is the appropriate dose or not remains a controversy.

In prospective studies, the relationship between pazopanib exposure and efficacy was investigated by evaluating the plasma pazopanib concentration rather than the dose intensity. Suttle et al. first reported that a pazopanib trough concentration (*C_t_*) >20.5 mg/mL was associated with clinical efficacy (i.e., PFS and tumor shrinkage) [8]. The threshold of *C_t_* was also meaningful in the adjuvant treatment of RCC [24]. In an investigation of the pazopanib exposure-response relationship in STS, however, the STS patients with *C_t_* > 20 mg/mL tended to have longer PFS, but a significant benefit was not proven (18.7 vs. 8.8 weeks, *n* = 26, *p* = 0.142) [9]. This result might be due to two reasons; the lack of power due to a small number of patients, and heterogeneity of pazopanib sensitivities by histology.

In the EORTC 62043 phase 2 trial and other retrospective reports, there were various differences in the pazopanib sensitivity and prognoses of STS patients by their histology; there were histological types with high sensitivity to pazopanib (such as leiomyosarcoma and alveolar soft part sarcoma), but some histological types, such as liposarcoma and the malignant peripheral nerve sheath tumor, were less sensitive to pazopanib treatment [11,20]. A predictive marker of pazopanib responses to STS has not been established, but there is a report that says the TP53 mutation might be useful as a predictive marker [25].

In our present retrospective investigation, we included the STS patients who achieved 12W-PFS as the pazopanib responders in the analyses with the intention to enroll patients who were considered to show a relatively stronger relationship between the pazopanib dose intensity and responses; as a result, a total of 78 patients (63%) among the 124 STS patients were included. The rate of pazopanib responders was higher than that reported in the EORTC 62043 phase 2 study (41%) [11]. Our analyses of all pazopanib-treated STS patients showed that RDI ≥80% throughout pazopanib treatment periods had significantly shorter PFS and OS; on the other hand, the landmark analyses showed that the 12W-RDI ≥80% pazopanib responders achieved significantly longer PFS, which may indicate that maintaining a high dose intensity, at least for the initial 12 weeks, could enable better and longer responses of STS patients to pazopanib. This paradoxical result might be derived from the heterogeneity of STS. Of the pazopanib responders, approx. 70% of the patients were able to maintain 12W-RDI ≥80%, and more than half of the patients were able to continue pazopanib without interruption and/or a dose reduction during the initial 12 weeks. Pazopanib responders with 12W-RDI ≥80% actually tended to keep relatively high RDI throughout the treatment periods.

The main reasons for pazopanib interruption and/or reduction in our patients were anorexia and liver function disorders; of them, liver function disorders, especially increased AST/ALT, were frequently observed in the 12W-RDI <80% pazopanib responders throughout the treatment period. Pazopanib is known to be metabolized in the liver, and hepatic toxicity is one of the important adverse events in pazopanib treatment [26]. Some genetic polymorphisms related to liver metabolism were reported to be associated with hepatic toxicity [27,28]. The detection of these polymorphisms prior to pazopanib treatment might help predict hepatic adverse events, enabling the reduction of dose intensity, as discussed and used in irinotecan treatment [29,30].

The limitations of our study were as follows: Because it was a retrospective study, follow-up periods or response evaluations were not strictly controlled as determined in advance, and the onset of each of the adverse events after the initial 12 weeks or change in patients’ quality of life (QOL) were not evaluated. The change in QOL during pazopanib treatment for STS patients was reported as an important topic [31], but we missed the evaluation of QOL in a fixed procedure in clinical practice. Some clinical data were also lacking, which could have been associated with RDI, such as geographical location and patients’ preferences, or details of prior chemotherapy regimens.

For the appropriate use of pazopanib, clinical trials monitoring pazopanib concentrations and trying to optimize the dose for each patient have been conducted [32,33,34]; however, the appropriate target concentration of pazopanib and the optimal monitoring periods are still under investigation. Our present finding that maintaining a high dose intensity during at least the initial 12 weeks may be beneficial for long-term responses among STS patients could be able to justify and support the motivation for continuing pazopanib at a high dose intensity for both patients and clinical practitioners. Our analyses also showed that the adverse events that differed in frequencies by dose intensity in pazopanib responders were liver function disorders. The evaluation of polymorphisms associated with liver metabolism might be the key for adjusting the pazopanib dose for STS patients as personalized medicine.

## Figures and Tables

**Figure 1 jcm-08-00060-f001:**
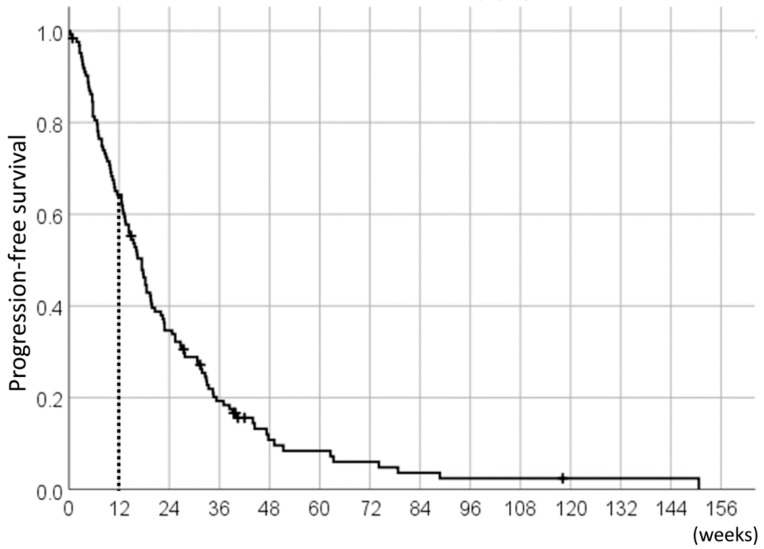
Progression-free survival (PFS) of the 124 pazopanib-treated soft tissue sarcoma (STS) patients.

**Figure 2 jcm-08-00060-f002:**
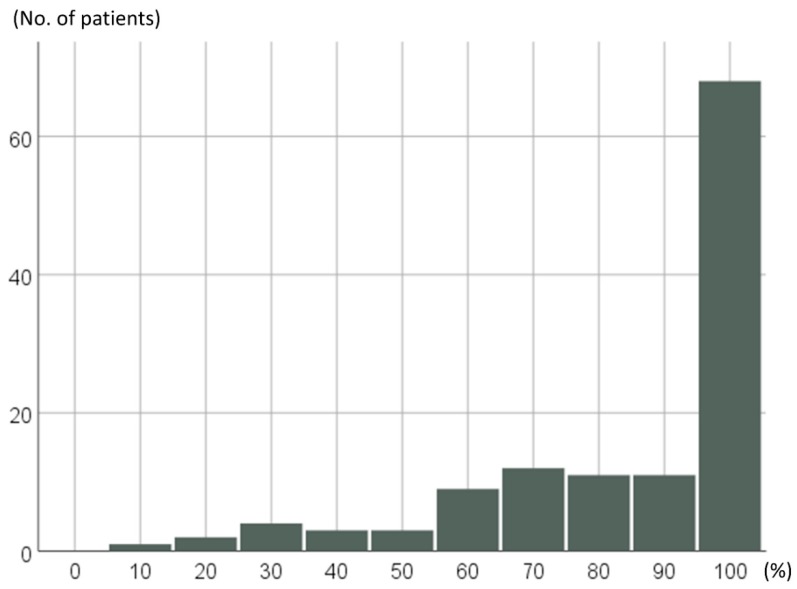
Distributions of relative dose intensity (RDI) separated by 10% in the 124 pazopanib-treated STS patients.

**Figure 3 jcm-08-00060-f003:**
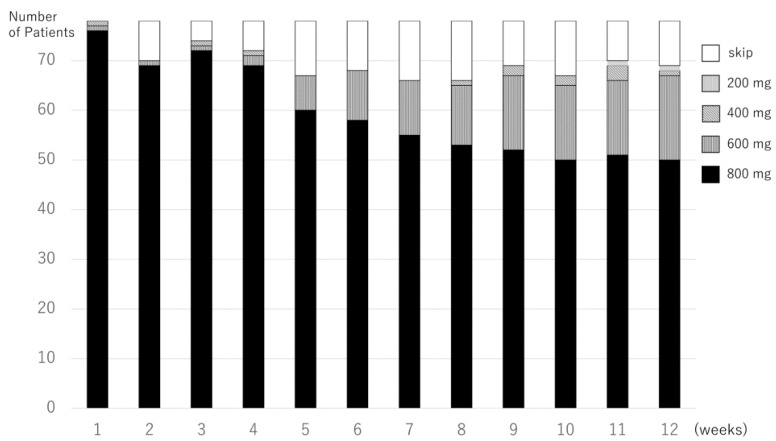
Pazopanib treatment doses in every week of the initial 12 weeks in the pazopanib responders.

**Figure 4 jcm-08-00060-f004:**
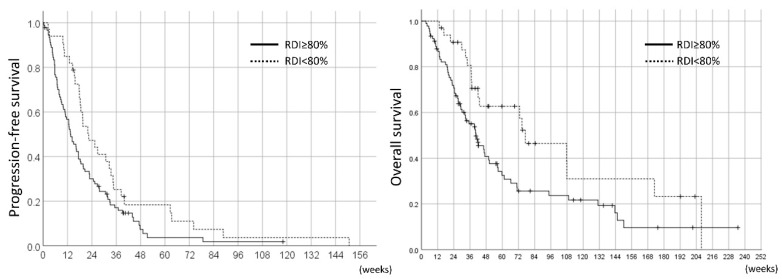
PFS and overall survival (OS) in the 124 pazopanib-treated STS patients by RDI throughout treatment periods.

**Figure 5 jcm-08-00060-f005:**
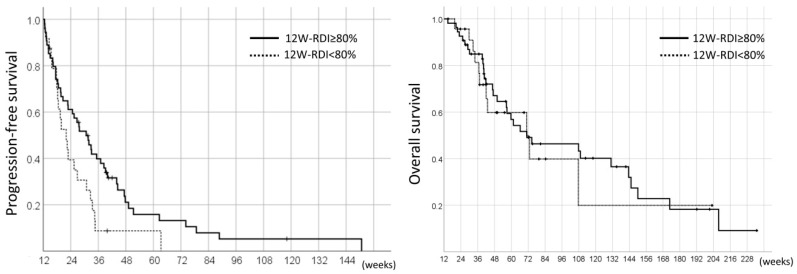
Landmark analysis of PFS and OS in the 78 pazopanib responders by RDI in the initial 12 weeks (12W-RDI).

**Table 1 jcm-08-00060-t001:** Patient characteristics.

Characteristic	All STS Patients Treated with Pazopanib (*n* = 124)	Pazopanib Responders (*n* = 78)
Age, years; median (range)	51 (19–85)	56 (19–85)
Male	57 (46%)	34 (44%)
Female	67 (54%)	44 (56%)
ECOG PS:		
0	78 (63%)	55 (71%)
1	43 (35%)	23 (29%)
2	3 (2%)	0
Primary lesion:		
Extremities	43 (35%)	30 (38%)
Non-extremities	81 (65%)	48 (62%)
Histology:		
Leiomyosarcoma	22 (18%)	15 (19%)
Synovial sarcoma	14 (11%)	11 (14%)
Liposarcoma	18 (15%)	10 (13%)
Other sarcoma	70 (56%)	42 (54%)
Prior chemotherapy:		
≤1 regimen	50 (40%)	35 (45%)
≥2 regimens	74 (60%)	43 (55%)

**Table 2 jcm-08-00060-t002:** Adverse events in the pazopanib responders.

Adverse Events	12W-RDI ≥ 80% (*n* = 54)		12W-RDI < 80% (*n* = 24)	
	All-Grade	≥ Grade 3	All-Grade	≥ Grade 3
Hematologic adverse events:	45 (83%)	12 (22%)	20 (83%)	7 (29%)
Leukocytopenia	21 (39%)	2 (4%)	11 (46%)	1 (4%)
Neutropenia	25 (46%)	4 (9%)	11 (46%)	3 (13%)
Lymphocytopenia	19 (35%)	8 (15%)	12 (50%)	3 (13%)
Anemia	14 (26%)	0	11 (46%)	0
Thrombocytopenia	19 (35%)	0	11 (46%)	0
Non-hematologic adverse events:	54 (100%)	12 (22%)	24 (100%)	11 (46%)
Diarrhea	31 (57%)	0	10 (42%)	0
Hypertension	29 (54%)	9 (17%)	15 (63%)	3 (13%)
Nausea/anorexia	28 (52%)	0	17 (31%)	0
Fatigue	26 (48%)	0	16 (67%)	0
Hand-foot-syndrome	20 (37%)	0	8 (33%)	0
Infection	10 (19%)	2 (4%)	3 (13%)	1 (4%)
Pneumothorax	7 (13%)	2 (4%)	0	0
Total bilirubin increase	17 (31%)	0	14 (58%)	0
AST/ALT increase	40 (74%)	1 (2%)	21 (88%)	8 (33%)
Serum creatinine increase	9 (17%)	0	7 (29%)	1 (4%)
Electrolytes disorder ^1^	25 (46%)	0	14 (58%)	2 (8%)
Hypothyroidism	30 (56%)	0	10 (42%)	0
Other adverse events ^2^	37 (69%)	4 (7%)	15 (63%)	1 (4%)

^1^ Electrolytes disorders included hypocalcinemia, hypercalcinemia, hyponatremia, hypokalemia, and hyperkalemia. The two patients with electrolytes disorders of Grade 3 or more observed in 12W-RDI <80% patients were both Grade 3 hypokalemia. ^2^ Other adverse events of Grade 3 or more observed in the 12W-RDI ≥80% patients were Grade 3 hyperglycemia, Grade 4 pulmonary embolism, Grade 3 ascites, and Grade 4 heart failure; that in the 12W-RDI <80% patient was Grade 3 mucositis.

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
