# Peer review of "Relative Dose Intensity of Induction-Phase Pazopanib Treatment of Soft Tissue Sarcoma: Its Relationship with Prognoses of Pazopanib Responders"

_jcm, 2019, doi:10.3390/jcm8010060_

Round 1
Reviewer 1 Report
In a retrospective studies on 78 out of 124 soft tissue sarcoma patients Kenji Nakano et al report the effect of pazopanib dosage on progression free survival (PFS). They used an interesting data set where they followed 78 pazopanib responders for PFS and concluded that majority of responders who were given current recommended dose of 800mg pazopanib for >12 weeks show better PFS as compared to the one whose dosage were reduced or withdrawn due to disease complications.
This study can definitely add to current knowledge gap in this field. I have few minor points only that can be incorporated in the discussion.
There is a major limitations in the field about quality-of-life data post 12 week regime. If possible, authors should add the information about quality-of-life of the patients who responded better vs who did not responded.
Additional factors such as geographical location and preference of patients, resource availability, patient choice such as route of administration choice etc that may have direct effect on data should be discussed.
Author Response
Thank you for reviewing our article.
We corrected the manuscript under your advice as follows:
- As you pointed, the lack of the quality-of-life data is a major limitaion of our study. We described about the limitation in Page 7, L204-205. Instead, we evaluated the RDI throughout the treatment periods in pazopanib-responders and showed them in Page 4, L121-123.
- We did not have clinical data about additional factors you pointed, so we referred to those factors as the limitaions of our study in Page 7, L206-207.
Please review and check the revised manuscript again.
Sincerely,
Kenji Nakano MD, PhD
Reviewer 2 Report
This retrospective study aims to explore a potential association between dose intensity on one side and response and tolerability on the other side in patients receiving pazopanib treatment for advanced or metastatic soft tissue sarcoma. While this is a clinically relevant research question, the study design has a number of shortcomings which lead to a potentially biased result and thus preclude a valid answer to the question.
My main concern is that dose intensity during early pazopanib treatment (in the present study defined as the first 12 weeks) is not necessarily causally related to response to later treatment. Pazopanib is a kinase inhibitor acting on several pathways which might be important for neoplastic activity. It usually acts fast and has an almost immediate effect on tumor metabolism. Therefore, I think it is indispensable to also show the possible association between dose intensity in the first 12 weeks and response in this time period. These data are unfortunately not part of the current analyses and should be added. In other words, you should analyze the entirety of patients who received pazopanib regarding dose intensity, response, and safety.
Another issue requiring clarification is the frequency of follow-up for response/progression and adverse events, as this has an direct effect on the results. Was there a fixed schedule for all patients? Were patients who had experienced adverse events or dose reduction followed up more frequently?
The legend of figure 2 reads: „The details of one of the 78 pazopanib-responders are shown in Figure 2.“. However, this does not match the figure, as it presents data not for a single patient, but for the entire cohort. It remains also unclear what the different dosages in the figure exactly relate to.
The introduction section could be written in a more concise manner.
The discussion section contains some "common sense" statements which are not really based on the results of the study but were already known before (e.g. "Our analyses also showed that the adverse events that should be monitored in pazopanib-responders were liver function disorders."). These could be omitted in order to shorten the discussion.
Author Response
Thank you for reviewing our article. We revised our manuscpirt under your advices as follows:
- Objective response rates and the timing of response shown of all patients were described in Page 3, L99-101. RDI of pazopanib-responders throughout the treatment periods were also described in Page 4, L121-123. Adverse events during all treatment periods were already shown in Table 2.
- The frequency of follow-up and radiological evaluations was described in Page 2, L70-72. Our study was retrospective, so the frequency of follow-up time was not strictly controlled; in the discussion (Page 7, L201-202), we referred to the limitaion.
- The explanation and figure legend of Figure 2 were corrected.
- We corrected the introduction section in the concise manner.
- We corrected the discussion section for deleting the "common sense" statements you pointed out.
I would like you to review our revised manuscript again.
Sincerely,
Kenji Nakano, MD, PhD
Round 2
Reviewer 2 Report
Unfortunately, you have not followed my recommendation to analyze the entirety of patients who received pazopanib regarding dose intensity, response, and safety in the revised version of your manuscript. To show a survival curve for PFS and present response rates without correlation to RDI is not sufficient.
Author Response
Thank you for reviewing our article again.
・We added RDI of all 124 patients throughout the pazopanib treatment in section 3.2 as Figure 1.
・The prognoses of all patients by RDI throughout the pazopanib treatment were also shown section 3.3 as Figure 4. Based on these data, the discussion was added in Page 7, L202-206.
Please review the revised article.
Sincerely,
Kenji Nakano, MD, PhD